# The Cost-Effectiveness of Expanding the UK Newborn Bloodspot Screening Programme to Include Five Additional Inborn Errors of Metabolism

**DOI:** 10.3390/ijns6040093

**Published:** 2020-11-20

**Authors:** Alice Bessey, James Chilcott, Abdullah Pandor, Suzy Paisley

**Affiliations:** School of Health and Related Research, the University of Sheffield, Sheffield S1 4DA, UK; j.b.chilcott@sheffield.ac.uk (J.C.); a.pandor@sheffield.ac.uk (A.P.); s.paisley@sheffield.ac.uk (S.P.)

**Keywords:** glutaric aciduria type 1, homocystinuria, isovaleric acidaemia, long-chain hydroxyacyl CoA dehydrogenase deficiency, maple syrup urine disease, inborn errors of metabolism, cost-effectiveness, economic, newborn screening, neonatal screening

## Abstract

Glutaric aciduria type 1, homocystinuria, isovaleric acidaemia, long-chain hydroxyacyl CoA dehydrogenase deficiency and maple syrup urine disease are all inborn errors of metabolism that can be detected through newborn bloodspot screening. This evaluation was undertaken in 2013 to provide evidence to the UK National Screening Committee for the cost-effectiveness of including these five conditions in the UK Newborn Bloodspot Screening Programme. A decision-tree model with lifetable estimates of outcomes was built with the model structure and parameterisation informed by a systematic review and expert clinical judgment. A National Health Service/Personal Social Services perspective was used, and lifetime costs and quality-adjusted life years (QALYs) were discounted at 1.5%. Uncertainty in the results was explored using expected value of perfect information analysis methods together with a sensitivity analysis using the screened incidence rate in the UK from 2014 to 2018. The model estimates that screening for all the conditions is more effective and cost saving when compared to not screening for each of the conditions, and the results were robust to the updated incidence rates. The key uncertainties included the sensitivity and specificity of the screening test and the estimated costs and QALYs.

## 1. Introduction

Maple syrup urine disease (MSUD), homocystinuria (HCU), isovaleric acidaemia (IVA), glutaric aciduria type 1 (GA1) and long-chain hydroxyacyl CoA dehydrogenase deficiency (LCHADD) are all rare inborn errors of metabolism (IEMs). If undiagnosed and untreated, these disorders can cause irreversible developmental delay, physical disability, neurological damage and death. Treatment consists of restricted diets and emergency regimens to prevent crises that lead to long-term harmful effects [1,2,3,4,5,6]. Screening for these conditions using tandem mass spectroscopy (TMS) to analyse heel prick bloodspots collected from newborns provides the possibility of early diagnosis and the initiation of preventive management with the potential to avoid or alleviate the associated adverse morbidity and mortality [7,8].

Within the United Kingdom (UK), the National Screening Committee (NSC) advises the government and the National Health Service (NHS) on which newborn screening tests should be offered. This advice is based on independent assessments of the evidence of the potential benefits, harms and cost-effectiveness of each screening test. From July 2012 to July 2013, the NHS National Institute for Health Research (NIHR) programme funded a pilot to evaluate the inclusion of the five conditions. The conditions were added to the then-current TMS panel of five conditions that were already screened for. This expanded newborn screening pilot involved six screening centres in England and screened around 440,000 babies, with 12 confirmed cases, during the 12-month pilot [9]. The NHS Research Ethics Service designated the Expanded Newborn Bloodspot Screening pilot and its evaluation as a Service Evaluation. 

The economic analysis presented here was conducted in parallel with the pilot study and evaluated the potential economic impact of adding each of the five conditions to the existing TMS newborn screening programme compared to not including them. Jansen’s 2017 review [10] highlighted the importance of ensuring that the evidence used in newborn screening policy making is in the public domain. This economic evaluation was included as part of the evidence considered by the NSC and informed their recommendation to include four (GA1, HCU, IVA and MSUD) of the five conditions in routine screening. Since the recommendation by the NSC in 2014, over 2.5 million babies have now been screened for the four conditions, which provides additional evidence for the specificity of the screening test and the screened incidence of the four conditions [11,12]. This allows the robustness of the original results to be evaluated against the new evidence and highlights some of the issues in predicting the cost-effectiveness of screening for rare conditions given a limited evidence base and weak country-specific data [11,12].

## 2. Materials and Methods

A cost-effectiveness decision tree model, shown in Figure 1, was built to estimate the impact of including each of the five conditions in the existing TMS NHS Newborn Bloodspot Screening Programme. The cost-effectiveness model provides the framework that enables evidence from a number of sources to be used together to estimate the costs and benefits of the novel intervention, in this case, over a lifetime horizon. Lifetables were used to estimate the lifetime cost and health outcomes for patients for each end node of the decision tree. A decision tree structure with lifetables captures the relevant outcomes without the complexity of a Markov Model. The lifetable approach is equivalent to a partitioned survival model in that it differentiates between the different disability levels. The outputs of the model include the expected incremental costs and health benefits of screening versus not screening, where the health benefits are expressed as quality-adjusted life years (QALYs) gained. The model takes an NHS/PSS (personal social services) perspective, whereby only costs that accrue to the NHS or PSS are included. Costs and benefits were discounted at the public health rate of 1.5% [13], and the costs were estimated at 2012 rates.

Systematic searches were undertaken for health-related quality of life, cost or economic evaluations, and disease natural history studies for each of the five conditions. Evidence from the systematic searches together with expert clinical opinion was used to estimate the parameters of the decision tree and the subsequent lifetables. The lifetables used to estimate the outcomes for each decision tree end node combined survival estimates with morbidity outcomes to estimate the associated QALYs, and the costs of managing both the conditions and the long-term complications arising from them. The overall cost and QALY parameters, as well as the other parameters used in the model, are shown in Table 1. More detail on the calculation of each parameter is given below. The description of the patient pathways and experience during the pilot was used to define the treatment pathways within the model.

The incidence for each of the five conditions was estimated based upon a systematic review of international studies in populations with and without screening [8,14]. The incidence data from the pilot were not directly included in the model due to the short duration of the pilot and the small numbers of detected cases expected; in total, the pilot identified 12 patients across the five conditions. Two potential biases are present within the published evidence on incidence. Firstly, the high rates of early mortality combined with the non-specific symptoms and lack of awareness of the conditions means that the pre-screening evidence may underestimate the true prevalence. Secondly, some of the conditions have a spectrum of severity, which means that screening may have the potential to identify individuals that may otherwise have remained asymptomatic. Studies were meta-analysed using a fixed-effect logit model within a Bayesian meta-analysis. For conditions where there was a marked difference in incidence between the screened and clinically detected populations, a lognormal bias adjustment was included within the model. For MSUD, HCU and LCHADD, the model estimates the birth incidence to be virtually equivalent in the screened and unscreened populations. For IVA and GA1, the data synthesis suggests that the screened incidence is much higher than the clinical incidence, which is in line with clinical expectations [15,16]. In the case of IVA, this is hypothesised to be due to the existence of a common, low-risk, potentially asymptomatic-phenotype mutation, c.932C>T. The sensitivity and specificity of the TMS device for each condition was estimated using a logit model within a Bayesian synthesis using data extracted from a systematic review using WinBUGS1.4 [8,17]. The incidence and the sensitivity and specificity estimates for each condition are shown in Table 1 [8,14]. 

Published case series were used to estimate survival with and without screening. Where evidence on survival following screen detection was not available, survival in asymptomatically diagnosed individuals, for instance, siblings, was used as a proxy. Expert judgment was used to supplement published evidence, where a best/worst-case scaling approach was used to elicit values from clinical experts. The survival rate projections in the screened and unscreened populations for each of the five conditions are shown in Figure 2, and a description of the data used for each of the conditions is given below. The marginal effects of treatment were taken directly from published case series or expert opinion, with normal survival assumed beyond the duration of the studies. In the case of IVA, LCHADD and MSUD, the survival impacts accrue in the first year of life. The impact on GA1 and HCU is longer lasting.

The mortality associated with symptomatically and screen-detected GA1 up to the age of 200 months was based on a published study [15], which compared outcomes in patients identified by newborn screening in Germany with a historical symptomatically identified cohort. In line with this study, the model assumes a normal life expectancy in the screen-detected cohort and for patients in the symptomatically detected cohort surviving beyond 200 months [15].

The mortality rate for symptomatically detected cases of HCU up to the age of 50 years was estimated from a published study [3]. While there is evidence that vascular events are reduced through the early commencement of dietary management [26], no direct evidence concerning the impact on mortality has been identified. Based on the literature [3,18] and expert judgment, a relative hazard of 0.20 (0.1–0.3) for screen-detected HCU mortality was used up to the age of 50 years. It was assumed that both cohorts would have a normal life expectancy after the age of 50 years. Based on these estimates, screening would increase survival at age 50 from 73% to 91%.

Evidence suggests that the mortality associated with symptomatically detected IVA occurs in the first two years of life, primarily in the neonatal period, with 25% of deaths occurring within the first seven days and only in those not carrying the low-risk c.932C>T mutation [2]. The model assumes that this same mortality rate occurs in the high-risk subgroup of the screen-detected population. It is also assumed that mortality after the first seven days is avoided in the screen-detected group and that survival is unaffected in the low-risk subgroup. It was estimated that 32% of patients were classed in the high-risk subgroup [16]. The model estimated a survival of approximately 95% in the screen-detected cohort of IVA patients and 80% in the symptomatically detected cohort.

Mortality in the screen- and symptomatically detected LCHADD over the first 8 years of life was taken from the data on patient outcomes from a published study [4]. The onset of symptoms associated with LCHADD can be early and sudden; there is, therefore, a continuing risk of mortality in the period between screening and treatment initiation following screen detection. The model estimates that survival in LCHADD patients would increase from approximately 60% at 8 years of age to 93%, with a normal life expectancy thereafter.

There was a lack of relevant data on MSUD survival in the literature, and therefore, expert judgement was used to estimate the mortality risk for the screen- and symptomatically detected cohorts. MSUD was split into three different categories: Classical, Intermediate (detectable), and Other. Expert elicitation was used to estimate the mortality rate in the first month of life for symptomatically detected and screen-detected cases. For classical MSUD, the mortality rate in the first month of life was estimated to be 26% (20%, 50% (min, max)) in the symptomatically detected cohort and 8% (5%, 10% (min, max)) in the screen-detected cohort. Normal survival was assumed in Intermediate and Other MSUD patients in both the screen- and symptomatically detected cohorts.

No studies directly assessing the quality-of-life impact of screening for any of the five conditions were identified. In order to estimate QALYs for the five conditions, published evidence and expert judgement concerning the short- and long-term health impacts and health and social care consequences of the conditions were used. The outcomes of screen- and symptomatically detected patients were split into normal, mild, moderate and severe developmental or neurological disability health states. The proportion of each condition falling into each health state is shown in Table 2. For GA1, LCHADD and MUSD, the QALY estimates for each of these health states were estimated using a visual analogue scale valuation of the EQ-5D in adults. For HCU and IVA, the utilities for each of the health states were estimated using the extended EQ-5D, which includes a cognition dimension (EQ-5D+C) in an effort to fully capture the impact of the condition [24]. The QALY or QALY decrements associated with each health state are shown in Table 2. It should be noted that the EQ-5D and also, by association, the EQ-5D+C are only assessed in the adult population and are known to have difficulties in application to a paediatric age group. However, whilst there are no value sets for a paediatric population, especially in the early and infant years [27], excluding the consideration of morbidity impacts would underestimate the potential impact of screening.

The morbidity in GA1 is caused by an encephalopathy crisis, which is most commonly associated with severe neurological impairment, including dystonia, speech problems, epilepsy and subdural bleeding. In a small proportion of cases, neurological disorders can arise without an encephalopathy crisis; these are characterised as either late or insidious onset cases and tend to be of a less severe form. The age distribution of the first encephalopathy crisis and the proportion of encephalopathy-free symptomatic patients who remain asymptomatic were taken from a published study [15], as was the proportion of screen-detected cases who experienced encephalopathy crises. The costs include the immediate impact of the encephalopathy crises together with costs for preventive hospitalisations; the lengths of hospitalisations for such episodes are estimated by clinical experts, with unit costs taken from routine data details, which are described in Appendix A. 

The evidence that was available from the literature on the morbidity associated with LCHADD was insufficient for directly estimating the quality-of-life and cost impacts for the model [1]. Expert elicitation was used to estimate the proportion of patients in each category of developmental delay for the purposes of estimating the utility values and the costs associated with the morbidity of the disorder.

Based on the literature, it was assumed that screen-detected cases of MSUD would have substantially fewer encephalopathy crises than those symptomatically detected. The decrease in encephalopathy crises would improve not only the mortality but also the morbidity associated with MSUD. There was a lack of data in the published studies on the impact on morbidity, and therefore, expert judgement was used to estimate the proportion of patients in each of the classic and intermediate categories of MSUD, together with the consequent impact on mild, moderate or severe disability due to developmental delay. The model assumes that all MSUD classic symptomatic cases and 10% of screen-detected cases involve an encephalopathy crisis [5].

The neurocognitive impairments associated with IVA were based on outcomes from a published study [2], where early diagnosis was defined as being in the first five weeks of life. The quality of life decrements associated with the differing levels of neurocognitive impairment were estimated from a methodological study [24] and matched with the levels used in the Grunert et al. study [2]. The learning disabilities category was assumed to be equivalent to the mid-level decrement (“some problems”) of the cognition attribute in the expanded EQ-5D+C; mild neurocognitive impairments were assumed to solely impact cognition, whilst severe impairments also impacted self-care and usual activities. The average quality-of-life decrements associated with the neurocognitive outcomes from early and late diagnoses were thus estimated as 0.04 and 0.18 QALYs, respectively. The costs associated with developmental delay in childhood were based on the costs identified for children with autism living in the family home; the adult costs were derived from a study reporting costs for a range of day services including living in the family home, staffed group homes or independent housing [28] and are described in more detail in Appendix A.

The model focuses on the improvements in quality of life through the avoidance of developmental delay consequent to the improved early management of HCU made possible by screen detection. It is assumed that the quality-of-life impacts associated with ocular problems [18] and vascular events are subsumed within the developmental delay decrement. Based on the literature [3,19], the majority of non-screen-detected cases have at least a mild developmental delay; the outcomes in the screened cohort are assumed to be the same as those for non-affected sibling controls, with a normal age-specific QALY for 85% of the cohort, with a decrement applied to 15% of the population due to poor dietary compliance. The quality-of-life and cost impacts of developmental delay are estimated as for IVA.

The marginal cost and resource estimates for expanding the TMS screening system were taken from the expanded newborn screening pilot. These included all elements of the process, including the midwife taking the bloodspot sample and providing information to parents, the identification of positive screen results and subsequent confirmatory testing and initial consultations, advice and dietary management for the period of the pilot follow-up. The cost per baby screened was estimated to be GBP 0.50 for the additional laboratory time, with an additional GBP 0.09 for the additional dietetic input, giving a total of GBP 413,000 per year of screening 700,000 babies.

The costs of the longer-term management of the conditions including routine appointments, blood tests and dietary management were estimated from the pilot study protocol, diet management advice developed for the screening pilot [22] and expert dietician input. The healthcare costs associated with managing the conditions, such as for appointments, routine tests and dietary supplements, were estimated using unit costs from a routine data sources [20,21,23] and are described in more detail in Appendix A. The health and social care costs arising from the complications of the conditions, such as encephalopathy crises, preventative admissions and the longer-term outcomes were also included, are described in more detail in Appendix A and are based on the outcomes described for each condition above and in Table 2. The discounted lifetime cost parameters associated with the conditions are shown in Table 1.

The cost and quality-of-life parameters were characterised by parametric distributions, described in Table 2. The baseline uncertainty analysis assumed a 15% coefficient of variation for all the long-term cost and quality-of-life estimates and 5% for all the short-term effects. A sensitivity analysis presented in Table 3 aimed at exploring some of the structural uncertainties in the model included a doubling of the uncertainty in all the long- and short-term parameters. The decision uncertainty was also examined with an expected value of perfect information (EVPI) analysis, which calculates the value in eliminating the uncertainty around the model parameter values and provides an estimate of the value for future research. The population EVPI was calculated for births over a five-year period.

Since the initial analysis was conducted in 2013, the UK has screened for the four recommended conditions (GA1, HCU, IVA and MSUD) in over 2.5 million babies. The screened incidence rate in the UK has proved to be lower for each condition than that predicted in the model based upon prior international published evidence. A deterministic sensitivity analysis was conducted that used the screened incidence rate and is shown in Table 1 [11,12]. In all the conditions apart from IVA, it was assumed that the screen- and symptomatically detected cases were the same. For IVA, it was assumed that the symptomatically detected incidence was the incidence of high-risk IVA and that screening also identified the mild cases.

The validation of the model included expert clinical input into the design and building of the model. In addition, the report to the NSC describing the modelling methods and results was subjected to a review via a formal stakeholder consultation as part of the NSC process and was reviewed by the NSC, whose membership includes clinical experts and health economists.

## 3. Results

Table 3 presents the estimated costs and quality-of-life effects of screening for each of the conditions compared to not screening for these conditions. The model estimates that screening for all the conditions is more effective and cost saving when compared to not screening for each of the conditions; therefore, screening is said to dominate not screening. The results also indicate that screening is very likely to be cost-effective at the commonly used UK thresholds for all the conditions apart from IVA, having a greater than 90% probability of being cost-effective at GBP 20,000 per QALY, and screening for all the conditions having a greater than 50% probability of being cost saving (see Appendix A). These economic results are driven by the projected improvements in survival and quality of life in screened patients, as shown in Figure 2 (and Appendix A), that also lead to substantial reductions in lifetime health and social care costs due to the avoidance of developmental problems in the early years of life. The overall costs, QALYs and life years for a screened and non-screened case for each condition are shown in Table 1.

Table 4 presents an analysis of the uncertainty by means of an EVPI analysis for the five conditions. The parameters giving rise to the greatest decision uncertainty are those associated with IVA, including the quality of life obtained in screen-detected children, the specificity of the test and the incidence in an unscreened population. The key uncertainties for MSUD are the sensitivity of the test and the costs of management in the screen-detected children; for HCU, the key uncertainty is the test specificity, whilst for GA1, the key uncertainties are the test specificity and incidence. For LCHADD, the uncertainties captured in the model do not impact the decision uncertainty. The value of further information for the other parameters is GBP 0 or close to GBP 0. As expected, doubling the uncertainty for each parameter increases the decision uncertainty and the value from collecting additional data. This is especially the case for IVA and, to a lesser extent, MSUD and GA1. Given the inherent uncertainty in the analysis, these results may better represent the value of collecting further evidence.

The use of the screening incidence rates in the UK since 2014 has resulted in lower incremental QALYs and cost benefits for all the conditions. However, screening is still estimated to be cost saving for all the conditions apart from IVA. The new incidence rates for IVA result in an incremental cost-effectiveness ratio (ICER) of GBP 776 per QALY, still well below the thresholds used in the UK.

## 4. Discussion

Expanding the existing TMS newborn screening programme to including screening for each of the five conditions considered here is estimated to both increase QALYs and reduce overall costs. The key economic uncertainties for all five conditions as outlined in the value of information analysis include the sensitivity and specificity of the screening test, the incidence of GA1 and IVA and the estimated QALYs for IVA. Based on the evidence regarding screening feasibility from the UK pilot of expanded newborn screening and the economic evidence provided by this analysis in 2014, the UK NSC recommended the inclusion of screening for GA1, HCY, IVA and MSUD in the bloodspot screening programme.

For MSUD, HCY and GA1, the economic value of resolving remaining uncertainties was low and, in most cases, centred on the screening test characteristics. There were much greater uncertainties for IVA, which revolved around the true clinical birth prevalence of IVA and the specificity of the test. The high over-detection estimated for IVA is most probably related to the prevalence of the 932C->T mutation, individuals with which are likely to remain asymptomatic and undetected without screening. The management costs for this subgroup in the form of regular clinical appointments were included in this analysis. However, no adverse quality-of-life impact was applied to these individuals to account for any disbenefits from the marginal impact of diagnosis and management. The pilot detected a number of low-risk IVA cases, and due to this, the NSC only recommended IVA screening with a lower test threshold than that initially used in the pilot in order to reduce potential over-detection [9].

Very little decision uncertainty was found in the economic modelling of LCHADD. However, there was concern that the modelling did not adequately capture the uncertainties surrounding the definitions, treatments and outcomes between LCHADD and the spectrum of conditions associated with mitochondrial trifunctional protein disorders. These problems with definitions make the parameterisation of the economic model difficult and subject to greater epistemic uncertainty than is captured within the probabilistic analysis. Prior to the pilot, the evidence suggested that very few cases would present in the neonatal period. However, during the pilot, LCHADD cases presented earlier than expected (<31 days), with a number presenting prior to the screening results being available. This limits the potential benefit from screening for pre-symptomatic diagnosis. Due to this, LCHADD was not recommended to be included in the expansion of the Newborn Screening Programme [9].

The sensitivity analysis suggests that the incidence rates experienced in the UK since 2014 would reduce the economic benefits of expanded screening but would not of themselves alter the decision to recommend screening, as the four conditions were still either cost saving or had ICERs well below the standard thresholds used in the UK. As with many of the parameters in the model, the original incidence rates were based on international rather than UK-specific data. International data are commonly used with rare conditions, given the small number of cases per country, and in cases where screening has already been implemented in other countries. This analysis demonstrates that incidence rates can vary markedly between countries, and estimates based upon unscreened and screened populations are both subject to bias [8,29,30]. This highlights the need for meta-analytical methods that capture the true uncertainty in predicted incidence rates when trying to generate early economic estimates of proposed screening interventions for rare conditions.

Furthermore, the screened incidence rates in the UK used here were based only on the number of screen-confirmed cases, as there was still no readily available evidence on the number of symptomatically detected cases available for the screening period. This issue highlights the importance of collecting ongoing evidence on the performance of screening programmes to allow re-evaluation and support screening system development and improvement.

The results from this study are in line with cost-effectiveness studies from other countries. Some studies found that screening dominated no screening for GA1 [31], HCU [32] and HCU and MSUD [33], while others found that screening for all conditions was found to be cost-effective compared to no screening [32,34,35,36,37,38]. There were key differences between the studies, which account for the differences in results. A major economic factor is whether the conditions were being added to an existing TMS screening panel with only the marginal cost of the test included in the analysis, as in this UK study, or a new TMS screening system was being established. Additionally, the incidence used and whether special education and social services costs were included impacted whether the programme was found to be cost saving or cost-effective. It is unsurprising that the cost of the test is a key factor in the cost-effectiveness of a screening programme, as a high cost per test, for all babies, can easily outweigh the benefits of reduced treatment costs and improved outcomes for the small number of children identified through screening.

Another key difference between the cost-effectiveness studies was whether they included QALYs or used life years as the main measure of effectiveness. Out of the ten studies, six used QALYs and one used disability-adjusted life years (DALYs), which were based either on expert opinion [34,38,39] or the use of proxy conditions from the literature [31,32,33,36]. This highlights the difficulty in identifying suitable evidence for the potential quality-of-life impact of screening for rare conditions [27]. Methodological research is therefore urgently required to develop approaches for measuring and valuing health effects in the paediatric population, especially in the early and infant years. Furthermore, condition-specific research is required to understand the ability of existing generic quality-of-life instruments to adequately capture the specific impact of the rare condition.

Estimating long-term outcomes can also be difficult for these rare disorders. Many studies focus on a single centre or a single country and therefore only contain small numbers of patients. This, combined with a lack of consistency in how outcomes are often reported, makes synthesis difficult. Larger registry studies can overcome some of these difficulties by promoting consistent outcome reporting. Further problems with estimating long-term outcomes arise from the need to rely on data from patients who have been diagnosed a number of years ago, reflecting historical management and diagnosis patterns. There is some evidence that relying on such data in models may slightly overestimate the benefit of screening [40,41,42,43]. One approach for addressing some of these uncertainties is the development of natural history models for conditions that allow for some of these potential biases in the data to be taken into account.

The generalisability of this study to other countries is limited by the use of an NHS/PSS perspective, UK-specific costs, UK pathways and management, and the perspective of an expansion of an existing TMS screening programme. While the disease incidence, outcomes and test characteristics in the basecase were derived from the international literature and would therefore be more applicable to other countries, caution must be shown, as incidence rates and potentially outcomes may vary between countries.

The issues with estimating the incidence and outcomes together with the case of LCHADD highlights some of the problems associated with estimating both the effectiveness and cost-effectiveness of screening programmes for rare disorders. Whilst modelling can assist in bringing evidence from a number of sources together and allow the examination of uncertainties, it still relies on the quality of the data and assumptions included within the model. Therefore, it is highly recommended that routine reporting and data collection infrastructure be put in place to enable ongoing data collection with the possibility of a further economic evaluation when sufficient data are collected to more accurately assess the effectiveness and cost-effectiveness of the programme.

## Figures and Tables

**Figure 1 IJNS-06-00093-f001:**
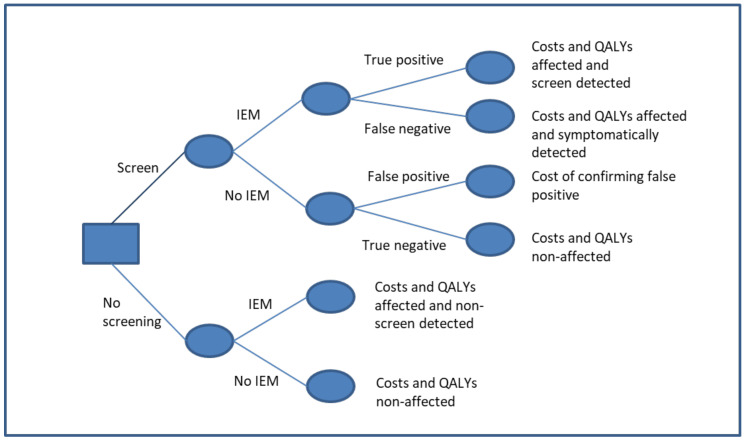
Model diagram. The figure represents the model structure.

**Figure 2 IJNS-06-00093-f002:**
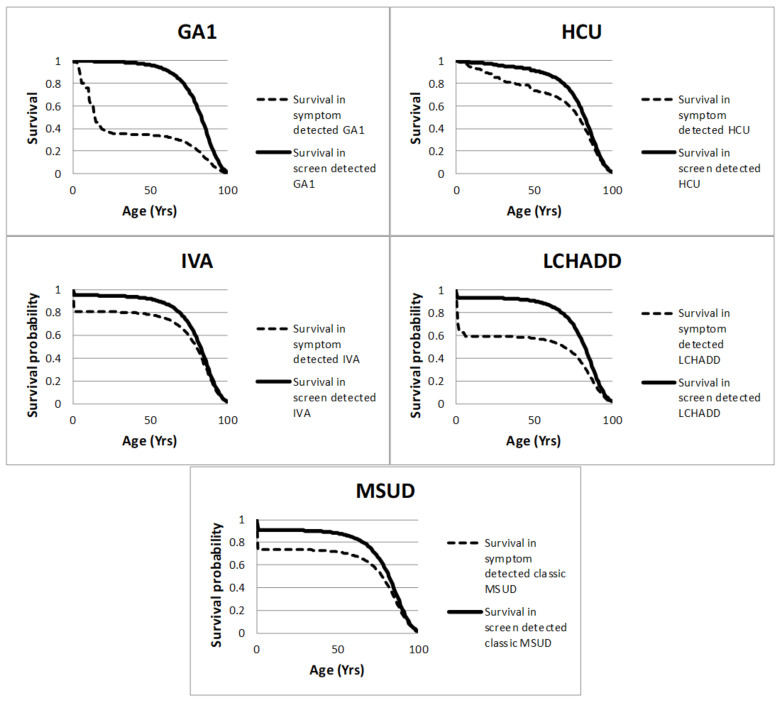
Modelled survival estimates in the screened and non-screened cohorts. This figure shows the modelled survival estimates for symptomatically and screen-detected cases against age for each of the five conditions.

**Table 1 IJNS-06-00093-t001:** Model Parameters.

Parameter		MSUD Mean (95% CI)	HCU Mean (95% CI)	IVA Mean (95% CI)	GA1 Mean (95% CI)	LCHADD Mean (95% CI)	Distribution	References
Cost of long-term health and social care impact of inborn errors	No Screening	£585,845 (£431k, £778k)	£704,459 (£519k, £936k)	£262,377 (£193k, £349k)	£549,529 (£405k, £730k)	£636,641 (£469k, £846k)	Lognormal	[2] (IVA) [3,18,19] (HCU) [5] (MSUD) [15] (GA1) [20,21] (All Conditions)
	Screening	£432,070 (£319k, £576k)	£82,193 (£61k, £109k)	£48,313 (£36k, £64k)	£170,644 (£126k, £227k)	£113,268 (£84k, £151k)		
	Incremental	−£153,775	−£622,267	−£214,064	−£378,885	−£523,373		
Cost of managing the IEM	No Screening	£445,933 (£327k, £589k)	£172,197 (£127k, £229k)	£171,859 (£127k, £229k)	£65,383 (£48k, £87k)	£81,900 (£60k, £109k)	Lognormal	[20,22,23]
	Screening	£531,328 (£393k, £707k)	£235,730 (£174k, £312k)	£69,704 (£52k, £92k)	£70,793 (£52k, £94k)	£56,578 (£42k, £75k)		
	Incremental	£85,394	£63,533	−£102,155	£5410	−£25,322		
Life-time QALYs	No Screening	14.17 (12.8, 15.6)	22.74 (20.6, 25.0)	29.90 (27.1, 32.9)	8.40 (7.6, 9.2)	17.80 (16.1, 19.6)	Lognormal	[2] (IVA) [3] (HCU), 5 (MSUD) [15] (GA1) [24] (HCU/IVA) [25] (GA1/LCHADD/MSUD)
	Screening	24.73 (23.3, 26.2)	38.40 (35.3, 40.9)	39.29 (36.4, 41.6)	39.47 (36.7, 41.7)	41.20 (39.2, 43.0)		
	Incremental	10.56	15.66	9.40	31.07	23.40		
Costs of confirmation testing in positive screening		£582 (£524, £638)	£475 (£428, £521)	£896 (£807, £983)	£1052 (£948, £1154)	£555 (£500, £609)	Normal	[21]
Screening test characteristics	Sensitivity	88.47% (16.6%, 100%)	93.26% (48.9%, 100%)	93.80% (56.5%, 100%)	90.72% (35.5%, 100%)	89.35% (38.3%, 100%)	Normal (logit)	[8]
	Specificity	99.99% (100.0%, 100%)	99.95% (99.7%, 100%)	99.99% (99.9%, 100%)	99.99% (99.9%, 100%)	100% (100%, 100%)	Normal (logit)	[8]
Incidence per 100,000 births	No Screening	0.73 (0.60, 0.87)	0.72 (0.52, 0.94)	0.30 (0.17, 0.47)	0.47 (0.26, 0.75)	0.55 (0.40, 0.68)	Normal (logit)	[8,15] (GA1), [16] (IVA) [8,14] (All Conditions)
	Screening	0.74 (0.60, 0.87)	0.74 (0.53, 0.95)	0.83 (0.69, 0.97)	1.02 (0.87, 1.17)	0.65 (0.52, 0.79)	Normal (logit)	
	Incremental	0.0003	0.02	0.52	0.56	0.10		
Sensitivity analysis: UK incidence per 100,000 births (2014–2018)	Screening and No Screening	0.5	0.347	0.154 (0.386 including mild)	0.309	N/A (not screened for)		[11,12]

IEM—inborn error of metabolism; QALYs—quality-adjusted life years; GA1—glutaric aciduria type 1; LCHADD—long-chain hydroxyacyl CoA dehydrogenase deficiency; MSUD—maple syrup urine disease; IVA—isovaleric acidaemia; HCU—homocystinuria; CI—credibility interval; k- thousand; -£ — incremental is below zero.

**Table 2 IJNS-06-00093-t002:** Morbidity distribution for each condition and quality-of-life estimates.

**Condition**	**Outcome**	**Normal**	**Mild Neurological or Psychiatric Disability**	**Moderate Neurological or Psychiatric Disability**	**Severe Neurological or Psychiatric Disability**	**References**
	QALY utility	1	0.721	0.503	0.075	[25]
GA1	Symptomatically detected	10%	0%	20%	70%	[15]
	Screen-detected	89.5%	0%	0%	10.5%	[15]
LCHADD	Symptomatically detected	12%	50%	30%	8%	Expert opinion
	Screen-detected	90%	6%	3%	1%	Expert opinion
MSUD Classic	Symptomatically detected	10%	50%	30%	10%	[5] Expert opinion
	Screen-detected	40%	40%	17.5%	2.5%	[5] Expert opinion
MSUD Intermediate	Symptomatic ally detected	35%	55%	10%	0%	[5] Expert opinion
	Screen-detected	100%	0%	0%	0%	[5] Expert opinion
Condition	Outcome	Normal	Learning disability	Mild developmental delay	Severe Developmental delay	
	QALY decrement	0	0.145	0.302	0.712	[24]
IVA	Symptomatic ally detected	33%	44%	11%	11%	[2]
	Screen-detected	82%	9%	9%	0%	[2]
HCU	Symptomatic ally detected	25%	0%	25%	50%	[3,19]
	Screen-detected	85%	0%	15%	0%	[3,19]

QALYs—quality-adjusted life years; GA1—glutaric aciduria type 1; LCHADD—long-chain hydroxyacyl CoA dehydrogenase deficiency; MSUD—maple syrup urine disease; IVA—isovaleric acidaemia; HCU—homocystinuria.

**Table 3 IJNS-06-00093-t003:** Results per baby screened.

	Condition	No Screening		Screening		Incremental		Cost-Effectiveness (ICER)	Probability Cost Saving/Dominating	Probability Cost-Effective			
		Cost	QALYs	Cost	QALYs	Cost	QALYs			£15,000 per QALY	£20,000 per QALY	£25,000 per QALY	£30,000 per QALY
Basecase analysis	MSUD	£7.58 (£5.65, £9.96)	41.79340 (40.13948, 43.44516)	£7.30 (£5.50, £9.44)	41.79347 (40.13953, 43.44524)	−£0.28 (−£2.42, £1.76)	0.000069 (0.000012, 0.000097)	Dominating	0.564	0.884	0.935	0.957	0.97
	HCU	£6.31 (£4.25, £9.01)	41.79146 (40.14842, 43.43173)	£2.98 (£1.89, £5.18)	41.79156 (40.14852, 43.43182)	−£3.33 (−£5.76, −£1.08)	0.000101 (0.000051, 0.000153)	Dominating	0.998	0.999	0.999	0.999	0.999
	IVA	£1.31 (£0.69, £2.12)	41.79356 (40.13962, 43.44532)	£1.20 (£0.90, £1.69)	41.79358 (40.13964, 43.44534)	−£0.10 (−£0.97, £0.63)	0.000014 (−0.000011, 0.000041)	Dominating	0.59	0.72	0.75	0.77	0.787
	GA1	£2.87 (£1.48, £4.89)	41.79344 (40.13949, 43.44511)	£2.72 (£2.05, £3.67)	41.79356 (40.13962, 43.44534)	−£0.15 (−£2.14, £1.37)	0.000120 (0.000034, 0.000218)	Dominating	0.542	0.933	0.967	0.981	0.99
	LCHADD	£3.94 (£2.65, £5.54)	41.79347 (40.13955, 43.44524)	£1.54 (£1.00, £3.04)	41.79358 (40.13966, 43.44536)	−£2.40 (−£4.04, −£0.76)	0.000114 (0.000046, 0.000158)	Dominating	0.997	1	1	1	1
Increased uncertainty sensitivity analysis	MSUD	£7.60 (£4.69, £11.73)	41.79340 (40.13947, 43.44516)	£7.30 (£4.64, £10.94)	41.79347 (40.13953, 43.44524)	−£0.30 (−£4.53, £3.83)	0.000069 (0.000012, 0.000105)	Dominating	0.512	0.726	0.778	0.832	0.87
	HCU	£6.31 (£3.51, £10.62)	41.79146 (40.14843, 43.43173)	£2.99 (£1.65, £5.38)	41.79156 (40.14852, 43.43181)	−£3.32 (−£7.47, −£0.44)	0.000101 (0.000031, 0.000164)	Dominating	0.991	0.999	0.999	0.999	0.999
	IVA	£1.31 (£0.62, £2.35)	41.79156 (40.14853, 43.43183)	£1.20 (£0.78, £1.87)	41.79157 (40.14841, 43.43183)	−£0.11 (−£1.21, £0.79)	0.000010 (−0.000049, 0.000048)	Dominating	0.568	0.649	0.665	0.676	0.686
	GA1	£2.86 (£1.25, £5.58)	41.79144 (40.14842, 43.43162)	£2.72 (£1.73, £4.23)	41.79155 (40.14838, 43.43182)	−£0.13 (−£2.81, £1.92)	0.000114 (0.000018, 0.000218)	Dominating	0.518	0.899	0.932	0.956	0.977
	LCHADD	£3.94 (£2.14, £6.70)	41.79146 (40.14844, 43.43172)	£1.53 (£0.87, £3.10)	41.79158 (40.14847, 43.43185)	−£2.40 (−£5.13, −£0.46)	0.000111 (0.000041, 0.000161)	Dominating	0.996	1	1	1	1
UK incidence rates: 2014–2018	MSUD	£5.18	41.79346	£5.03	41.79351	−£0.15	0.000047	Dominating	N/A	N/A	N/A	N/A	N/A
	HCU	£3.05	41.79153	£1.61	41.79158	−£1.44	0.000051	Dominating	N/A	N/A	N/A	N/A	N/A
	IVA	£0.67	41.79358	£0.68	41.79359	£0.01	0.000008	£776	N/A	N/A	N/A	N/A	N/A
	GA1	£1.90	41.79350	£1.07	41.79358	−£0.83	0.000087	Dominating	N/A	N/A	N/A	N/A	N/A

GA1—glutaric aciduria type 1; LCHADD—long-chain hydroxyacyl CoA dehydrogenase deficiency; MSUD—maple syrup urine disease; IVA—isovaleric acidaemia; HCU—homocystinuria; ICER—incremental cost-effectiveness ratio; -£ — incremental is below zero; N/A—not applicable, the analysis was run as a deterministic sensitivity analysis, and therefore, the probability of cost-effectiveness was not calculated.

**Table 4 IJNS-06-00093-t004:** Expected value of perfect information results.

Basecase ANALYSIS						
Parameter		GA1	HCU	IVA	LCHADD	MSUD
Cost IEM	No screening	£0	£0	£0	£0	£0
	Screening	£0	£0	£0	£0	£206
Cost Management	No screening	£0	£0	£0	£0	£0
	Screening	£0	£0	£13	£0	£1229
QALYs	No screening	£0	£0	£0	£0	£0
	Screening	£0	£0	£17,483	£0	£0
Screening Test	Sensitivity	£202	£336	£1421	£21	£3469
	Specificity	£77,769	£1753	£48,069	£0	£0
	Cost	£0	£0	£0	£0	£0
Incidence	Screening	£0	£0	£0	£0	£0
	No screening	£3078	£0	£162,786	£0	£0
Increased uncertainty sensitivity analysis						
Parameter		GA1	HCU	IVA	LCHADD	MSUD
Cost IEM	No screening	£0	£0	£3218	£0	£5985
	Screening	£255	£0	£3936	£0	£64,737
Cost Management	No screening	£0	£0	£6	£0	£19
	Screening	£0	£0	£16,837	£0	£122,995
QALYs	No screening	£0	£0	£19,594	£0	£0
	Screening	£5793	£0	£319,310	£0	£0
Screening Test	Sensitivity	£349	£59	£2756	£54	£6262
	Specificity	£8628	£1716	£56,129	£0	£0
	Cost	£0	£0	£0	£0	£0
Incidence	Screening	£0	£0	£894	£0	£0
	No screening	£9569	£0	£260,748	£0	£0

GA1—glutaric aciduria type 1; LCHADD—long-chain hydroxyacyl CoA dehydrogenase deficiency; MSUD—maple syrup urine disease; IVA—isovaleric acidaemia; HCU—homocystinuria; IEM—inborn error of metabolism; QALYs—quality-adjusted life years.

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
