# Peer review of "The Cost-Effectiveness of Expanding the UK Newborn Bloodspot Screening Programme to Include Five Additional Inborn Errors of Metabolism"

_2409-515X, 2020, doi:10.3390/ijns6040093_

Round 1

Reviewer 1 Report

The authors summarize a report developed for the NHS intended to assess the cost-effectiveness of newborn screening for five rare conditions. This is a difficult problem, due to the high uncertainty and lack of evidence on the development of these diseases. However, the authors make a smart use of modelling and combine the scarce information with expert knowledge by using appropriate methods. The article is well written, methods are rigorously applied, and conclusions are supported by the results.

I would only suggest some minor corrections:

  • Table 2 is difficult to understand. Please consider an alternative design or enriching the explanation.
  • I think I can guess that the lifetime costs for each condition shown in Table 1 are computed by combining the survival curves in Figure 2 with the costs shown in the supplementary material. Maybe this is not completely clear in the text of the manuscript. The same happens to QALYs. Please consider a more explicit explanation on how these sources of information are combined.
  • Did you perform sensitivity analyses on the key parameters detected by the EVPI? Apart from estimating EVPI, it would be very useful to provide some insights on the effect of changes of those parameters on the final results.

Reviewer 2 Report

This is an economic evaluation which was conducted in parallel with a 12-month pilot to evaluate the inclusion of five conditions (GA1, HCU ,IVA, LCHADD and MSUD) in England six years ago. However, given that there is already a recommendation from UK NSC back in May 2014 to have four of these five conditions included in the newborn screening programme, I have a few concerns below.

Major revision

  1. Given how the expanded UK NSC screening programme with these conditions has been running in England from early 2015, why hasn't the authors updated their analysis to make it more relevant to the readers now?
  2. Year of this economic evaluation is missing in the Abstract; this may mislead readers into thinking there was a new program evaluated when this is actually an economic evaluation performed as part of the recommendation considered by UK NSC back in May 2014.
  3. Please explain in the Introduction why this article is still relevant now even though the recommendation has been made six years ago.
  4. Can a brief description of the pilot be added please?
  5. What is the objective of this study - is it (a) comparing the then-existing newborn screening programme with the then-existing newborn screening programme plus the five additional conditions or (b) comparing no screening at all with then-existing newborn screening programme plus the five additional conditions? If it's (b), why isn't it compared to the standard of care = the then-existing screening programme?
  6. Is it the public service perspective (stated in Abstract) or NHS/PSS perspective (stated in Materials and Methods) that the authors are looking at? Can the terminology used be standardised?
  7. What is the source of parameters for all the tables; can they be stated within the tables for transparency please?
  8. Since there was a pilot for 12 months, was data from the pilot used in the first 12 months of the model?
  9. Why was a decision tree chosen over a Markov model?
  10. Please explain how this model was validated.
  11. Where are the probability parameters of the model?
  12. How was expert opinion elicited?
  13. What were the extrapolation assumptions used in this model?
  14. The following relates to the QALY of neonates:
    1. What is the reason for using EQ-5D+C?
    2. Is it validated for neonates?
    3. Is there a UK tariff for EQ-5D+C?
    4. How is QALY obtained from EQ-5D+C?
  15. Can the authors explain why a threshold of £25,000 per QALY was adopted?
  16. As IJNS is not a health economics journal, please explain "dominating" and "dominated" before using the terminology.
  17. Where are the uncertainty and sensitivity analyses of the cost-effectiveness analysis to assess methodological uncertainty?
  18. What is the probability that the expanded screening program will be cost-effective at different cost-effectiveness thresholds (£20,000 per QALY to £30,000 per QALY as recommended by NICE and £15,000 per QALY to reflect recent trends in healthcare decision-making)?
  19. How is the probabilistic analysis conducted? What are the values of the parameters examined? What distribution was assumed for each parameter?
  20. Can the 95% confidence interval be presented with the results please?
  21. What is the generalisability of this study?
  22. The ethics statement is missing, please add whether ethics approval was obtained or a statement stating it is not relevant.

Minor revision

  1. Abbreviation EQ-5D+C was not defined in the first place.

Round 2

Reviewer 2 Report

All the suggestions have been adequately addressed by the authors.

A minor change regarding stating the reasons for the specific type of decision analytical model which they have used in the manuscript would be good. Although it's explained in their letter of response, it's not in the manuscript. It's good practice to explain it as stated in item #15 of the CHEERS checklist.

Author Response

We have now included the text from the reviewers response into the manuscript (lines 73-75).

"A decision tree structure with lifetables captures the relevant outcomes without the complexity of a Markov Model. The lifetable approach is equivalent to a partitioned survival model in that it differentiates between the different disability levels."